# Towards A Unified Agent with Foundation Models

**Norman Di Palo**[*]
Imperial College London
London, UK
`n.di-palo20@imperial.ac.uk`

**Arunkumar Byravan**
DeepMind
London, UK

**Leonard Hasenclever**
DeepMind
London, UK

**Markus Wulfmeier**
DeepMind
London, UK

**Nicolas Heess**
DeepMind
London, UK

**Martin Riedmiller**
DeepMind
London, UK

## Abstract

Language Models and Vision Language Models have recently demonstrated unprecedented capabilities in terms of understanding human intentions, reasoning, scene understanding, and planning-like behaviour, in text form, among many others. In this work, we investigate how to embed and leverage such abilities in Reinforcement Learning (RL) agents. We design a framework that uses language as the core reasoning tool, exploring how this enables an agent to tackle a series of fundamental RL challenges, such as efficient exploration, reusing experience data, scheduling skills, and learning from observations, which traditionally require separate, vertically designed algorithms. We test our method on a sparse-reward simulated robotic manipulation environment, where a robot needs to stack a set of objects. We demonstrate substantial performance improvements over baselines in exploration efficiency and ability to reuse data from offline datasets, and illustrate how to reuse learned skills to solve novel tasks or imitate videos of human experts.

## 1 Introduction

In recent years, the literature has seen a series of remarkable Deep Learning (DL) success stories (3), with breakthroughs particularly in the fields of Natural Language Processing (4; 19; 8; 29) and Computer Vision (2; 25; 36; 37). Albeit different in modalities, these results share a common structure: large neural networks, often Transformers (46), trained on enormous web-scale datasets (39; 19) using self-supervised learning methods (19; 6). While simple in structure, this recipe led to the development of surprisingly effective Large Language Models (LLMs) (4), able to process and generate text with outstanding human-like capability, Vision Transformers (ViTs) (25; 13) able to extract meaningful representations from images and videos with no supervision (6; 18), and Vision-Language Models (VLMs) (2; 36; 28), that can bridge those data modalities describing visual inputs in language, or transforming language descriptions into visual outputs. The size and abilities of these models led the community to coin the term Foundation Models (3), suggesting how these models can be used as the backbone for downstream applications involving a variety of input modalities.

This led us to the following question: can we leverage the performance and capabilities of (Vision) Language Models to design more efficient and general reinforcement learning agents? After being trained on web-scaled textual and visual data, the literature has observed the emergence of common sense reasoning, proposing and sequencing sub-goals, visual understanding, and other properties in these models (19; 4; 8; 29). These are all fundamental characteristics for agents that need to interact with and learn from environments, but that can take an impractical amount of time to emerge tabula rasa from trial and error. Foundation Models are generally trained using thousands of PetaFLOPS/s-days (19), reutilising this computational effort to bootstrap learning, as well as leveraging their emergent properties in novel scenarios, is critical.

---

[*]work done during an internship at DeepMind.

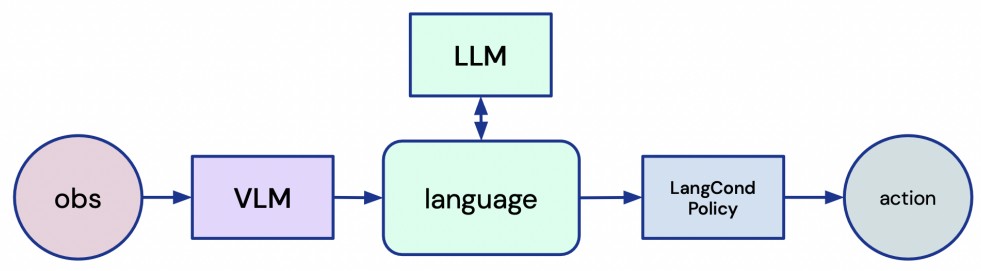

Figure 1: A high-level illustration of our framework.

Motivated by this idea, we design a framework that puts language at the core of an RL robotic agent, particularly in the context of learning from scratch. Our core contribution and finding is the following: we show that this framework, which leverages LLMs and VLMs, can tackle a series of fundamental problems in RL settings, such as **1) efficiently exploring** sparse-reward environments, **2) re-using collected data** to bootstrap the learning of new tasks sequentially, **3) scheduling learned skills** to solve novel tasks and **4) learning from observation** of expert agents. In the recent literature, these tasks need different, specifically designed algorithms to be tackled individually, while our proposed framework provides a more unified approach.

## 2 RELATED WORK

Over the past few years, scaling the parameter count of models and the size and diversity of training datasets led to unprecedented capabilities in (Vision) Language Models (4; 19; 2; 19; 8). This in turn led to several applications leveraging these models within agents that interact with the world. Prior work has used LLMs and VLMs together with RL agents in simulated environments (12; 44), but they rely on collecting large amounts of demonstrations for training agents. Instead, we focus on the problem of learning RL agents from scratch and leverage LLMs and VLMs to accelerate progress.

Prior work has also looked at leveraging LLMs and VLMs for robotics applications; particularly (1; 21; 50; 20) leveraged LLMs for planning sub-goals in the context of long-horizon tasks together with VLMs for scene understanding and summarization. These sub-goals can then be grounded into actions through language-conditioned policies (22; 30). While most of these works focus on deploying and scheduling already learned skills through LLMs, albeit in the real world, our work focuses on an RL system that learns such behaviours from scratch, highlighting the benefits that these models bring to exploration, transfer and experience reuse.

Several methods have been proposed to tackle sparse-reward tasks, either through curriculum learning (43; 51; 31; 16), intrinsic motivation (17; 35), or hierarchical decomposition (32; 27). We demonstrate how LLMs can generate learning curriculums *zero-shot*, without any additional learning or finetuning, and VLMs can automatically provide rewards for these sub-goals, greatly improving learning speed.

Related work has also looked at reusing large datasets of robotic experience by learning a reward model for the new tasks at hand (5). However, numerous human annotations of desired rewards need to be gathered for each new task. Instead, as reported in concurrent related work (48), we show successful relabeling of past experience leveraging VLMs which can be finetuned with small amounts of data from the target domain.

(15) is the most similar method to our work: they propose an interplay between LLMs and VLMs to learn sparse-reward tasks in Minecraft (23; 24). However, there are some notable differences: they use a vast internet dataset of videos, posts and tutorials to finetune their models, while we demonstrate that it is possible to effectively finetune a VLM with as few as 1000 datapoints, and use off-the-shelf LLMs; additionally, we also investigate and experiment how this framework can be used for data reuse and transfer and learning from observation, besides exploration and skills scheduling, proposing a more unified approach to some core challenges in reinforcement learning.

## 3 PRELIMINARIES

We use the simulated robotic environment from Lee et al. (26) modelled with the MuJoCo physics simulator (45) for our experiments: a robot arm interacts with an environment composed of a red, a blue and a green object in a basket. We formalise it as a Markov Decision Process (MDP): the observation space $\mathcal{O}$ is composed of $128 \times 128 \times 3$ RGB images coming from two cameras fixed to the edges of the basket. The state space $\mathcal{S}$ represents the 3D position of the objects and the end-effector. The robot is controlled through position control: the action space $\mathcal{A}$ is composed of an $x, y$ position, that we reach using the known inverse kinematics of the robot, where the robot arm can either pick or place an object, inspired by (49; 40). The agent receives a language description of the task $\mathcal{T}$ to solve, which can have two forms: either "*Stack X on top of Y*", where *X* and *Y* are taken from {"*the red object*", "*the green object*", "*the blue object*" } without replacement, or "Stack all three objects", that we also call *Triple Stack*.

A positive reward of $+1$ is provided if the episode is successful, while a reward of $0$ is given in any other case. We define the *sparseness* of a task as the average number of environment steps needed, when executing random actions, to solve the task and receive a single reward. With the MDP design we adopt, stacking two objects has a sparseness of $10^3$. Stacking all three objects has a sparseness of more than $10^6$ as measured by evaluating trajectories from a random policy.

## 4 A FRAMEWORK FOR LANGUAGE-CENTRIC AGENTS

The goal of this work is to investigate the use of Foundation Models (3), pre-trained on vast image and text datasets, to design a more general and unified RL robotic agent. We propose a framework that augments from-scratch RL agents with the ability to use the outstanding abilities of LLMs and VLMs to reason about their environment, their task, and the actions to take entirely through language.

To do so, the agent first needs to map visual inputs to text descriptions. Secondly, we need to prompt an LLM with such textual descriptions and a description of the task to provide language instructions to the agent. Finally, the agent needs to ground the output of the LLM into actions.

**Bridging Vision and Language using VLMs:** To describe the visual inputs taken from the RGB cameras (Sec. 3) in language form, we use **CLIP**, a large, contrastive visual-language model (36). CLIP is composed of an image-encoder $\phi_I$ and a text-encoder $\phi_T$, trained on a vast dataset of noisily paired images and text descriptions, that we also refer to as captions. Each encoder outputs a 128-dimensional embedding vector: embeddings of images and matching text descriptions are optimised to have large cosine similarity. To produce a language description of an image from the environment, the agent feeds an observation $o_t$ to $\phi_I$ and a possible caption $l_n$ to $\phi_T$ (Fig. 2). We compute the dot product between the embedding vectors and considers the description correct if the result is larger than $\gamma$, a hyperparameter ($\gamma = 0.8$ in our experiments, see Appendix for more details). As

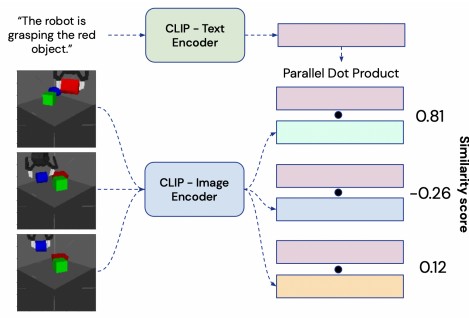

Figure 2: An illustration of CLIP computing the similarity, as dot product, between observations and text descriptions.

we focus on robotic stacking tasks, the descriptions are in the form "*The robot is grasping X*" or "*The X is on top of Y*", where *X* and *Y* are taken from {"*the red object*", "*the green object*", "*the blue object*" } without replacement. We finetune CLIP on a small amount of data from the simulated stacking domain; more details on how this works and analysis on data needs for finetuning are provided in the appendix.

**Reasoning through Language with LLMs:** Language Models take as input a prompt in the form of language and produce language as output by autoregressively computing the probability distribution of the next token and sampling from this distribution. In our setup, the goal of LLMs is to take a text instruction that represents the task at hand (e.g. "*Stack the red object on the blue object*"), and generate a set of sub-goals for the robot to solve. We use **FLAN-T5** (10), an LLM finetuned on datasets of language instructions. A qualitative analysis we performed showed that it performed slightly better than LLMs not finetuned on instructions.

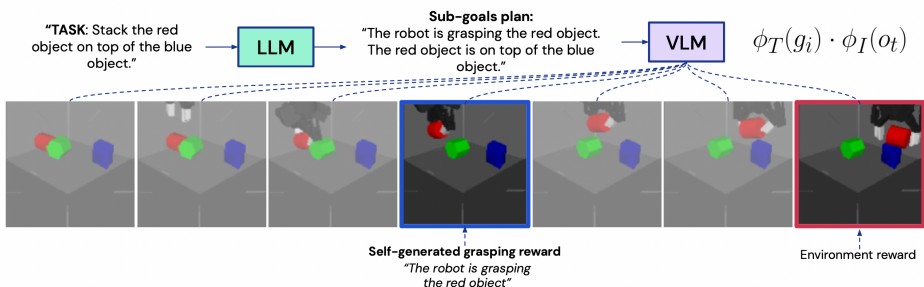

Figure 3: The VLM can act as an internal reward model by comparing language goals proposed by the LLM to the collected observations.

The extraordinary in-context learning capabilities of these LLMs allowed us to use them off-the-shelf (4; 34), without the need for in-domain finetuning, and guide their behaviour by providing as few as two examples of task instruction and desired language outputs: we describe the environment setting, asking the LLM to find sub-goals that would lead to solving a proposed task, providing two examples of such tasks and relative sub-goals decomposition. With that, the LLM was able to emulate the desired behaviour, not only in content, but also in the formatting of the output language which allowed for efficient parsing. In the Appendix we provide a more detailed description of the prompts we use and the behaviour of the LLMs.

**Grounding Instructions into Actions:** The language goals provided by the LLMs are then grounded into actions using a language-conditioned policy network. This network, parameterized as a Transformer (46), takes an embedding of the language sub-goal and the state of the MDP including object and robot end-effector positions as input, each represented as a different vector, and outputs an action for the robot to execute. This network is trained from scratch within an RL loop as we describe below.

**Collect & Infer Learning Paradigm:** Our agent learns from interaction with the environment through a method inspired by the Collect & Infer paradigm (38). During the Collect phase, the agent interacts with the environment and collects data in the form of states, observations, actions and current goal as $(s_t, o_t, a_t, g_i)$, predicting actions through its policy network, $f_\theta(s_t, g_i) \rightarrow a_t$. After each episode, the agent uses the VLM to infer if any sub-goals have been encountered in the collected data, extracting additional rewards, as we explain in more detail later. If the episode ends with a reward, or if any reward is provided by the VLM, the agent stores the episode data until the reward timestep $[(s_0, o_0, a_0, g_i), \ldots, (s_{T_r-1}, o_{T_r-1}, a_{T_r-1}, g_i)]$ in an experience buffer. We illustrate this pipeline in Fig. 4 (Left). These steps are executed by $N$ distributed, parallel agents, that collect data into the same experience buffer ($N = 1000$ in our work). During the Infer phase, we train the policy through Behavioural Cloning on this experience buffer after each agent has completed an episode, hence every $N$ total episodes, implementing a form of Self-Imitation on successful episodes (33; 14; 7). The updated weights of the policy are then shared with all the distributed agents and the process repeats.

## 5 APPLICATIONS AND RESULTS

We described the building blocks that compose our framework. The use of language as the core of the agent provides a unified framework to tackle a series of fundamental challenges in RL. In the following sections, we will investigate each of those contributions, focusing on **exploration**, **reusing past experience data**, **scheduling and reusing skills** and **learning from observation**. The overall framework is also described in Algorithm 1.

### 5.1 EXPLORATION - CURRICULUM GENERATION THROUGH LANGUAGE

RL benefits substantially from carefully crafted, dense rewards (5). However, the presence of dense rewards is rare in many real-world environments. Robotic agents need to be able to learn a wide range of tasks in complex environments, but engineering dense reward functions becomes prohibitively time-consuming as the number of tasks grows. Efficient and general exploration is therefore imperative to overcome these challenges and scale RL.

A wide variety of methods have been developed over the years to tackle exploration of sparse-reward environments (43; 51; 31; 16; 17; 35; 32; 27). Many propose decomposing a long-horizon task into

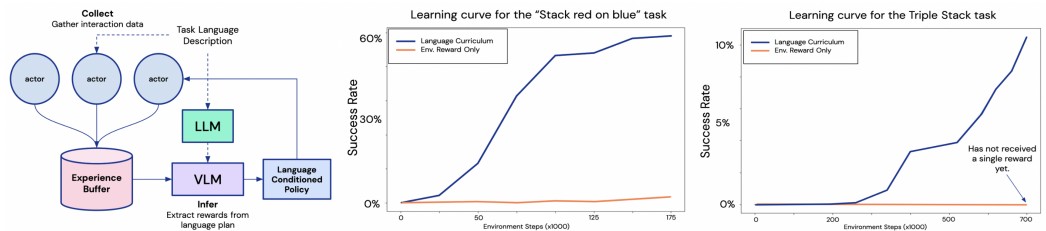

Figure 4: **Left:** Illustration of our Collect & Infer pipeline. **Middle, Right**: Learning curves of our framework and a baseline in the *Stack Red on Blue* and *Triple Stack* tasks.

shorter, easier to learn tasks, through curriculum generation and learning. Usually, these methods need to learn to decompose tasks from scratch, hindering overall learning efficiency. We demonstrate how an RL agent leveraging LLMs can take advantage of a curriculum of text sub-goals that are generated *without any past environment interaction*.

To guide exploration, the agent provides the task description $\mathcal{T}_n$ to the LLM, instructing it to decompose the task into shorter-horizon sub-goals, effectively generating a curriculum of goals $g_{0:G}$ in text form [1]. The agent selects actions as $f_\theta(s_t, \mathcal{T}_n) \to a_t$. While the environment provides a reward only if $\mathcal{T}_n$ is solved, the VLM is deployed to act as an additional, less sparse reward model: given the observations $o_{0:T}$ collected during the episode and all the text sub-goals $g_{0:G}$ proposed by the LLM, it verifies if any of the sub-goals were solved at any step.

We consider an observation $o_t$ to represent a completion state for a sub-goal $g_i$ if $\phi_T(g_i) \cdot \phi_I(o_t) > \gamma$. In that case, the agent adds $[(s_0, o_0, a_0, \mathcal{T}_n), \ldots, (s_{t-1}, o_{t-1}, a_{t-1}, \mathcal{T}_n)]$ to our experience buffer. The process is illustrated in Fig. 3, 11 (in the Appendix).

**Results on Stack *X* on *Y* and Triple Stack.** We compare our framework to a baseline agent that learns only through environment rewards in Fig. 4. The learning curves clearly illustrate how our method is substantially more efficient than the baseline on all the tasks. Noticeably, our agent's learning curve rapidly grows in the *Triple Stack* task, while the baseline agent still has to receive a single reward, due to the sparseness of the task being $10^6$. We provide a visual example of the extracted sub-goals and rewards in the Appendix.

---

**Algorithm 1** Language-Centric Agent

1: **// Training time:**
2: **for** task in tasks **do**
3:     subgoals = LLM(task) *//find text subgoals given task description*
4:     exp_buffer.append( VLM(offline_buffer, subgoals)) *//extract successful eps from offline buff. collected in past tasks* (**Sec. 5.2**)

5:     **for** $ep$ in episodes **do**
6:         (**Sec. 5.1**)
7:         $E \leftarrow [s_{0:T}, o_{0:T}, a_{0:T}, g_i]$ *//collect ep. trajectory*
8:         $r \leftarrow$ collect final reward
9:         $r_{internal} \leftarrow$ VLM($E$, subgoals) *//extract additional rewards for subgoals*
10:         **if** $r$ or $r_{internal}$ **then**
11:             exp_buffer.append($E_{0:T_r}$) *//Add timesteps until reward*
12:         **if** $ep\%N == 0$ **then**
13:             $\theta \leftarrow$ BC(episode_buffer) *//train agent with BC every $N$ eps*
14: **// Test time:**
15: Receive *text_instruction* or *video_demo*
16: **if** *text_instruction* **then**
17:     subgoals = LLM(*text_instruction*) (**Sec. 5.3**)
18: **else if** *video_demo* **then**
    subgoals = VLM(*video_demo*) (**Sec. 5.4**)
19: execute(subgoals) (**Sec. 5.3**)

---

These results suggest something noteworthy: we can compare the sparseness of the tasks with the number of steps needed to reach a certain success rate, as in Fig. 5. We train our method also on the *Grasp the Red Object* task, the easiest of the three, with sparseness in the order of $10^1$. We can see that, under our framework, the number of steps needed grows more slowly than the sparseness of the task. This is a particularly important result, as generally the opposite is true in Reinforcement Learning (35).

---

[1]For example, the LLM decomposes "*Stack the red object on the blue object*" into the following sub-goals: ["*The robot is grasping the red object*", "*The red object is on top of the blue object*"]

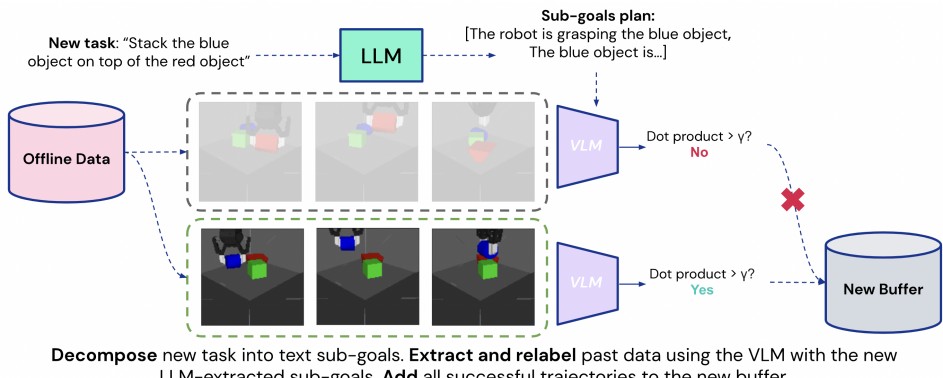

**Decompose** new task into text sub-goals. **Extract and relabel** past data using the VLM with the new LLM-extracted sub-goals. **Add** all successful trajectories to the new buffer.

Figure 6: Our framework can reutilise offline data collected on other tasks, extracting successful trajectories for the new task at hand, bootstrapping policy learning.

This slower growth, enabled by the increase in the amount of sub-goals proposed by the LLM as the task becomes sparser, suggests that our framework can scale to even harder tasks and make them tractable, assuming sub-goals can be encountered with a uniform-like distribution at any point during exploration. Additionally, unlike prior approaches that need carefully crafted intrinsic rewards or other exploration bonuses our approach can directly leverage prior knowledge from LLMs and VLMs to generate a semantically meaningful curriculum for exploration, thereby paving the way for general agents that explore in a self-motivated manner even in sparse-reward environments.

Figure 5: With our framework, the number of steps needed to reach a certain success rate grows more slowly than the sparseness of the task.

## 5.2 EXTRACT AND TRANSFER - EFFICIENT SEQUENTIAL TASKS LEARNING BY REUSING OFFLINE DATA

When interacting with their environments, our agents should be able to learn a series of tasks over time, reusing the prior collected data to bootstrap learning on any new task instead of starting *tabula rasa*. This is a fundamental ability to scale up RL systems that learn from experience. Recent work has proposed techniques to adapt task-agnostic offline datasets to new tasks, but they can require laborious human annotations and learning of reward models (5; 47; 9).

We leverage our language based framework to showcase bootstrapping based on the agent's past experience. We train three tasks in sequence: *Stack the red object on the blue object*, *Stack the blue object on the green object*, and *Stack the green object on the red object*, that we call $[\mathcal{T}_{R,B}, \mathcal{T}_{B,G}, \mathcal{T}_{G,R}]$. The intuition is simple: while exploring to solve, for example, $\mathcal{T}_{R,B}$, it is likely that the agent had solved other related tasks, like $\mathcal{T}_{B,G}$ or $\mathcal{T}_{G,R}$, either completely or partially. The agent should therefore be able to extract these examples when trying to solve the new tasks, in order not to start from scratch, but reuse all the exploration data gathered for previous tasks.

As discussed in Sec. 4, our agent gathers an experience buffer of interaction data. We now equip the agent with two different buffers: a *lifelong buffer*, or *offline buffer*, where the agent stores each episode of interaction data, and continues expanding it task after task. Then, the agent has a *new task buffer*, re-initialised at the beginning of each new task, that is filled, as in Sec. 5.1, with trajectories that result in a reward, either external or internally provided by the VLM using LLM text sub-goals (Fig. 3). The policy network is optimised using the *new task buffer*.

Differently from before however, while the first task, $\mathcal{T}_{R,B}$, is learned from scratch, the agent reuses the data collected during task $n$ to bootstrap the learning of the next task $n+1$. The LLM decomposes $\mathcal{T}_{n+1}$ into text sub-goals $[g_0, \ldots, g_{L-1}]$. The agent then extracts from the *lifelong/offline buffer* each

stored episode $\mathcal{E}_n = [(s_{0:T,n}, o_{0:T,n}, a_{0:T,n})]$. It then takes each episode's observation $o_{t,n}$ and uses the VLM to compute dot-products score between all image observations and all text sub-goals as $\phi_T(g_l) \cdot \phi_I(o_t)$. If the score is larger than the threshold $\gamma$ the agent adds all the episode's timesteps up to $t$, $[(s_{0:t,n}, o_{0:t,n}, a_{0:t,n})]$ to the *new task buffer*. The process is illustrated in Fig. 6.

This procedure is repeated for each new task at the beginning of training. Following this procedure, the agent does not start learning new tasks *tabula rasa*: at the beginning of task $\mathcal{T}_n$, the current experience buffer is filled with episodes useful to learn the task extracted from $\mathcal{T}_{0:n}$. When $n$ increases, the amount of data extracted from $\mathcal{T}_{0:n}$ increases as well, speeding up learning.

**Results on Experience Reuse for Sequential Tasks Learning.** The agent applies this method to learn $[\mathcal{T}_{R,B}, \mathcal{T}_{B,G}, \mathcal{T}_{G,R}]$ in succession. At the beginning of each new task we re-initialise the policy weights: our goal is to investigate the ability of our framework to extract and re-use data, therefore we isolate and eliminate effects that could be due to network generalisation.

We plot how many interaction steps the agent needs to take in the environment to reach 50% success rate on each new task in Fig. 7. Our experiments clearly illustrate the effectiveness of our technique in reusing data collected for previous tasks, improving the learning efficiency of new tasks.

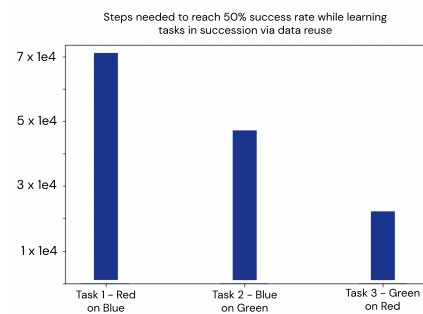

Figure 7: In our experiments, the agent can learn task $n+1$ faster than task $n$ by reusing past experience data.

These results suggest that our framework can be employed to unlock lifelong learning capabilities in robotic agents: the more tasks are learned in succession, the faster the next one is learned. This can be particularly beneficial when deploying agents in open-ended environments, particularly in the real world; by leveraging data across its lifetime the agent has encountered it should be able to learn novel tasks far faster than learning purely from scratch.

## 5.3 SCHEDULING AND REUSING LEARNED SKILLS

We described how our framework enables the agent with the ability to efficiently explore and learn to solve sparse-reward tasks, and to reuse and transfer data for lifelong learning.

Using its language-conditioned policy (Sec. 4), the agent can thus learn a series of $M$ skills, described as a language goal $g_{0:M}$ (e.g. "*The green object is on top of the red object*" or "*The robot is grasping the blue object*").

Our framework allows the agent to schedule and reuse the $M$ skills it has learned to solve novel tasks, beyond what the agent encountered during training. The paradigm follows the same steps we encountered in the previous sections: a command like *Stack the green object on top of the red object*

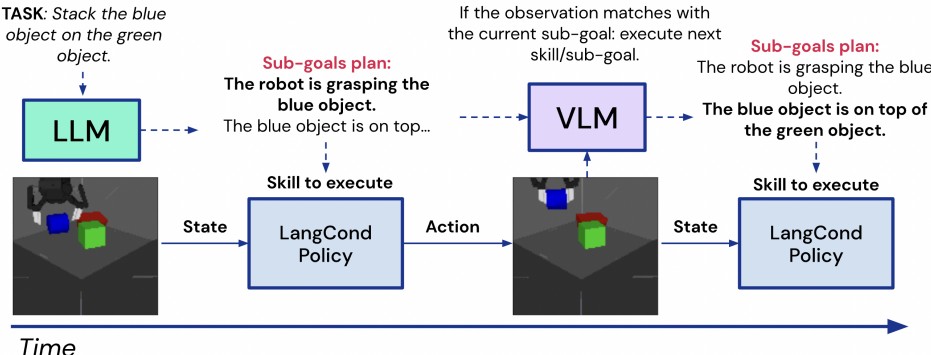

Figure 8: Our framework can break down a task into a list of skills using the LLM, and execute each skill until the VLM predicts that its sub-goal has been reached.

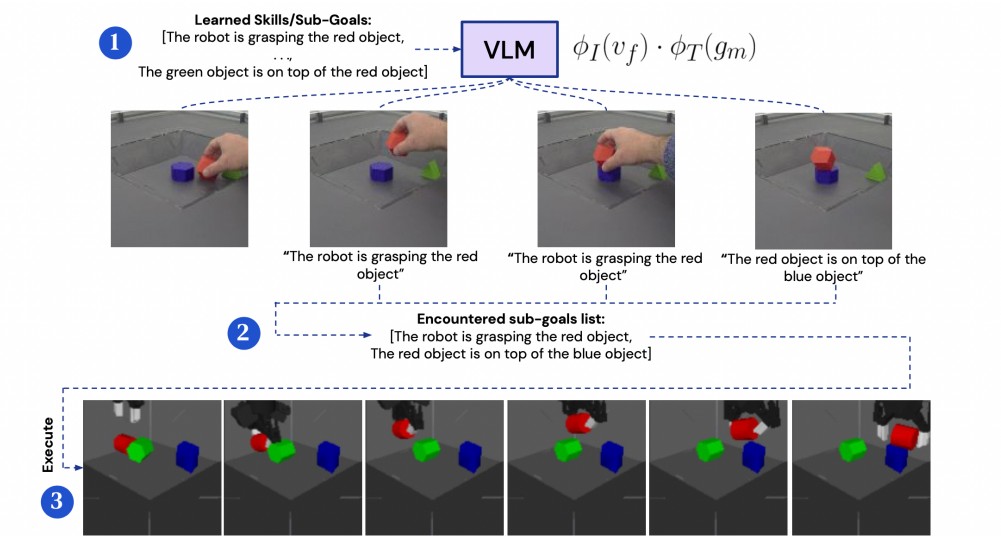

Figure 9: An illustration of the agent learning from observation using our framework.

or *Stack the red on the blue and then the green on the red* is fed to the LLM, which is prompted to decompose it into a list of shorter-horizon goals, $g_{0:N}$. The agent can then ground these into actions using the policy network as $f_\theta(s_t, g_n) \to a_t$.

When executing the $n$-th skill, the VLM computes at each timestep if $\phi_T(g_n) \cdot \phi_I(o_t) > \gamma$, thus checking if the goal of the skill has been reached in the current observation. In that case, the agent starts executing the $n + 1$-th skill, unless the task is solved.

## 5.4 Learning from Observation: Mapping Videos to Skills

Learning from observing an external agent is a desirable ability for general agents, but this often requires specifically designed algorithms and models (42; 11; 52). Our agent can be conditioned on a *video* of an expert performing the task, enabling *one-shot learning from observation*. In our tests, the agent takes a video of a human stacking the objects with their hand. The video is divided into $F$ frames, $v_{0:F}$. The agent then uses the VLM, paired with the $M$ textual description of the learned skills, expressed as sub-goals $g_{0:M}$, to detect what sub-goals the expert trajectory encountered as follows: (1) the agent embeds each learned skill/sub-goal through $\phi_T(g_m)$ and each video frame through $\phi_I(v_f)$ and compute the dot product between each pair. (2) it lists all the sub-goals that obtain a similarity larger than $\gamma$, collecting the chronological list of sub-goals the expert encountered during the trajectory. (3) It executes the list of sub-goals as described in Fig. 8.

Despite being finetuned only on images from the MuJoCo simulation (Sec. 4), the VLM was able to accurately predict the correct text-image correspondences on real-world images depicting both a robot or a human arm. Notice also how we still refer to it as "*the robot*" in the captions (Fig. 9), but the VLM generalises to a human hand regardless.

## 6 Conclusion

We propose a framework that puts language at the core of an agent. Through a series of experiments, we demonstrate how this framework provides a more unified approach with respect to the current literature to tackle a series of core RL challenges that would normally require separate algorithms and models: **1)** exploring in sparse-reward tasks **2)** reusing experience data to bootstrap learning of new skills **3)** scheduling learned skills to solve novel tasks and **4)** learning from observing expert agents. These initial results suggest that leveraging foundation models can lead to general RL algorithms that can tackle a variety of problems with improved efficiency and generality. By leveraging the prior knowledge within these models we can design better robotic agents that are capable of solving challenging tasks directly in the real world. We provide a list of current limitations and future work in the Appendix.

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

## 7 APPENDIX

### 7.1 FINETUNING CLIP ON IN-DOMAIN DATA

In our experiments, the dot products between the embeddings of possible captions and of an RGB observation from our environment $y = \phi_I(o_t) \cdot \phi_T(l_i)$ were often uninformative: correct and wrong pairs obtained very similar scores, and varied too little in range. Our goal is to set a threshold $\gamma$ to recognise correct and wrong descriptions given an image: therefore we need a larger difference in score. To tackle this, we collect a dataset of image observations with various configurations of the objects and the corresponding language descriptions using an automated annotator based on the MuJoCo state of the simulation to finetune CLIP with in-domain data. The plot on the right provides an analysis of our findings: precision and recall tend to increase logarithmically with the dataset size. The key takeaway message is that, although CLIP is trained on around $10^8$ images, just $10^3$ in-domain pairs are enough to improve its performance on our tasks.

In our case, a high precision is more desirable than high recall: the former indicates that positive rewards are not noisy, while the opposite may disrupt the learning process. A lower recall indicates that the model may not be able to

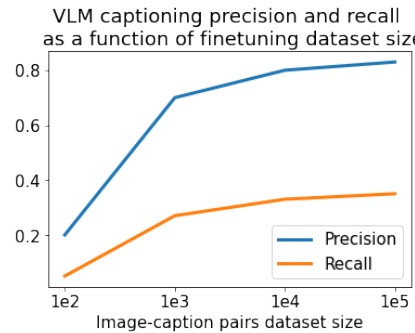

Figure 10: Captioning precision and recall of finetuned CLIP as a function of the dataset size. The logarithmic trend suggests that around $10^3$ image-caption pairs unlock sufficient performance. Values obtained with $\gamma = 0.8$.

correctly identify all successful trajectories, but this simply translate in the need for more episodes to learn, and does not disrupt the learning process. We found a value of $\gamma = 0.8$ to be the best performing choice after finetuning.

### 7.2 CURRENT LIMITATIONS AND FUTURE WORK

**1)** In our current implementation, we use a simplified input and output space for the policies, namely the *state space* of the MDP - i.e. the positions of the objects and the end-effector as provided by the MuJoCo simulator - and a pick and place *action space*, as described in Sec. 3, where the policy can output a $x, y$ position for the robot to either pick and place. This choice was adopted to have faster experiments iteration and therefore be able to focus our search on the main contribution of the paper: the interplay with the LLM and the VLM. Nevertheless, the recent literature has demonstrated that a wide range of robotics tasks can be executed through this action space formulation (49; 40).

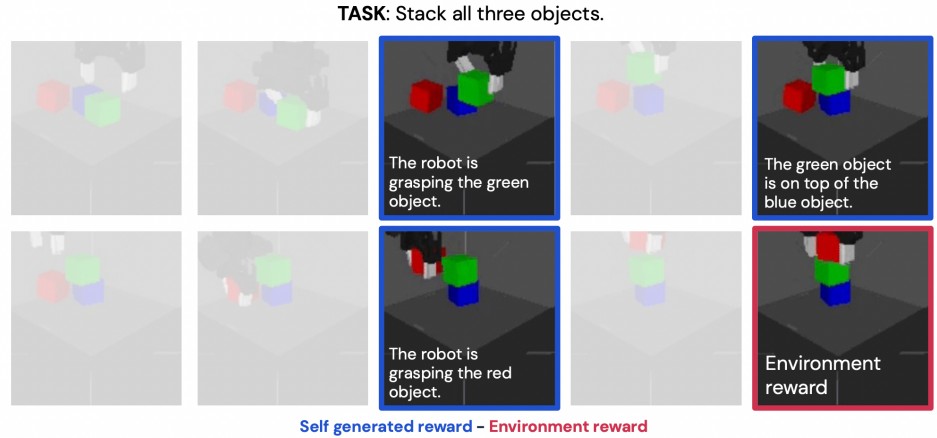

Figure 11: Autonomously identifying sub-goals and corresponding rewards becomes especially important when tasks become prohibitively sparse, like *Triple Stack*.

Imagine you are a robot arm interacting with an environment. In front of you there are a red object, a blue object, and a green object. When receiving a TASK, describe a set of SUBGOALS that would make you solve the task.

TASK: Stack the red object on top of the green object.
SUBGOALS: 1) The robot is grasping the red object.
2) The red object is on top of the green object.

TASK: Grasp the red object.
SUBGOALS: 1) The robot is grasping the red object.

TASK: Stack the blue object on top of the red object.
SUBGOALS: **1) The robot is grasping the blue object.**
**2) The blue object is on top of the red object.**

TASK: Stack all three objects.
SUBGOALS: **1) The robot is grasping the red object.**
**2) The red object is on top of the green object.**
**3) The robot is grasping the blue object.**
**4) The blue object is on top of the red object.**

Figure 12: An example of the prompt we used to condition the LLM, and its outputs. Normal text: user inserted text, **bold text:** LLM outputs.

Many works from the current literature (26; 41; 5; 15) demonstrate that, in order for the policy to scale to *image observations* as input and *end-effector velocities* as output, the model only needs more data, and therefore interaction time. As our goal was demonstrating the *relative performance improvements* brought by our method, our choice of MDP design does not reduce the generality of our findings. Our results will most likely translate also to models that use images as inputs, albeit with the need for more data.

**2)** We finetune CLIP on in-domain data, using the same objects we then use for the tasks. In future work, we plan to perform a larger scale finetuning of CLIP on more objects, possibly leaving out the object we actually use for the tasks, therefore also investigating the VLM capabilities to generalise to inter-class objects. At the moment, this was out of the scope of this work, as it would have required a considerable additional amount of computation and time.

**3)** We train and test our environment only in simulation: we plan to test the framework also on real-world environments, as our results suggest that 1) we can finetune CLIP with data from simulation and it generalises to real images (Sec. 5.4), therefore we can avoid expensive human annotations 2) the framework allows for efficient learning of even sparse tasks *from scratch* (Sec. 5.1), suggesting the applicability of our method to the real-world, where collecting robot experience is substantially more time expensive.

### 7.3 PROMPTS AND OUTPUTS OF THE LLM

In Fig. 12 we show the prompt we used to allow in-context learning of the behaviour we expect from the LLM (34). With just two examples and a general description of the setting and its task, the LLM can generalise to novel combinations of objects and even novel, less well-defined tasks, like "*Stack all three objects*", outputting coherent sub-goals.

