# OpenReview forum: "Towards A Unified Agent with Foundation Models"
_ICLR.cc/2023/Workshop/RRL — RRL 2023 Oral_

### Official Review · Reviewer_C3JQ · 2023-02-23

**Rating:** 4
**Confidence:** 4

**Review:**

**Description of the method**

This manuscript presents an RL method which leverages pre-trained language and vision models to condition the policy with natural language instructions.  A language model is used to generate subgoals descriptions from task descriptions.  Then a vision language model is given the textual sub-goal and the observation frames.  The VLM is then used to infer if any subgoals have been solved from the collected data.
CLIP is used to encode images and text from the frames and textual descriptions of the frames (with some finetuning on the actual environment).  InstructGPT is used as the language model to generate subgoals.  Trajectories with solved subgoals are added to the set of experience.  The main component here is that subgoals being solved provides an additional signal to identify successful trajectories.

This method can solve a highly sparse environment reward and hierarchical tasks such as a block stacking robotic task against a baseline which achieves no reward.  This work then adds an additional “new task buffer”, which is re-initialized at the start of each new task and is used to train the policy with sequntially learned tasks. Samples from the standard buffer are added to the new task buffer for sufficiently similar interactions, hence allowing reuse of the offline data.  Additionally, they present an HRL method where the agent learns a set of skills where the language model decomposes the task into a set of short horizon goals.  They experiment with utilizing real world images of humans stacking objects with their hands during pre-training and then finetune the model on images from the simulation.  This serves as an improvement in zero shot learning from observations.

**Review**

This paper is relatively well written, but the method is highly complex composed on many parts.  The figures serve well in describing the method in a concise way. I still found many of the textual descriptions hard to follow.

The work only utilizes a singular stacking task but presents substantial improvements and variations of the method.  I cannot clearly find the performance results for sections 5.3 and 5.4. I also cannot easily find details of the policy network.

I recommend accepting this paper since the utilization of the LLM and VLM for subgoal generation and checking is quite novel, highly interesting to the workshop and the performance improvements on the single yet difficult task are substantial.    The baseline cannot solve the task with any reward.

---

### Official Review · Reviewer_qoaU · 2023-02-26
**Strong accept - Innovative use of VLMs and effective application of hindsight relabeling make the paper highly relevant to the workshop topic.**

**Rating:** 4
**Confidence:** 5

**Review:**

*Summary*:

The paper presents a new way to improve reinforcement learning (RL) by combining Large Language Models (LLMs) and Visual-Language Models (VLMs). Their approach involves using an LLM to break down a task description into smaller sub-tasks. These sub-tasks are then used as inputs for a transformer-based agent to interact with the environment. After each round of interaction, a VLM is used to determine which observations completed each sub-task successfully. This information is used to create a dense reward function that helps the agent learn more efficiently.

The authors of the paper argue that while using LLMs to convert tasks into sub-tasks has been explored in previous studies, their research introduces two new innovative ideas. The first is combining these sub-tasks with a VLM to create a dense reward function, thus allowing the agent to learn more effectively. The second innovation is the use of this hindsight relabeling capability to reuse data collected from previous tasks or previously learned skills.

*Pros*:

- The paper is well-written and easy to follow and understand.
- The basic idea of using VLM and sub-tasks as a way to create a dense reward function is novel and promising. To the best of my knowledge there are only two papers that proposed a similar idea, the MineCLIP from MineDojo and ELLM (https://arxiv.org/pdf/2302.06692.pdf). Since both of them came out recently, I consider them a concurrent work.
- The paper shows how fundamental models can be effectively used in the RL framework, which is a topic that is currently of great interest and relevance to the research community.
- A significant portion of the experiments conducted by the authors focuses on how to use the relabeling capabilities to facilitate the reuse of prior calculations, either through stored trajectories or learned skills. Since this is the main topic of the workshop, I believe that this paper is a strong, suitable match for the workshop.

*Cons/Questions*:

- My main concern is the ability to use the proposed algorithm in more complex and rich environments. Prior work (Say-Can, for example) showed that in a complex environment, the sub-tasks an LLM output not always relevant to the scenario at hand, and one should worry about this alignment problem. In this paper, this problem was dealt with using a carefully chosen and engineered prompt. Since this task is simple, the prompt and two examples were enough for the LLM to be able to generate sub-tasks successfully. In richer tasks, I'm not sure that this approach is feasible. The same goes for the use of VLM to compare the observations to sub-tasks. If in such simple environment, the approach didn't work without fine-tuning of CLIP, what would happen in complex environments?
- The proposed algorithm clearly demonstrate strong capabilities to solve the task at hand. However, it is not clear enough which of the design choices the authors made is the main enabler for that. I believe that more ablations will help strengthen the paper. Specifically, I would propose comparing the Collect & Infer approach to conventional RL (still with the dense reward produced by the VLM and LLM). Another proposed ablation is to check your algorithm against other baselines that tried to deal with the sparse reward problem. Currently, the only comparison is only to a plain environment reward which is a required but not very convincing baseline .
- Why the agent's architecture is a transformer? Is there a reason to use a non-markovian agent for this problem?
- In my personal opinion, the title and abstract of the paper may overstate its contributions. While the use of VLMs to generate a dense reward function is an innovative approach, other aspects of the paper, such as using LLMs to break down tasks into sub-tasks and using hindsight reward relabeling for new tasks, have already been explored in previous research.